# Electrostatically controlled spin polarization in Graphene-CrSBr magnetic proximity heterostructures

Boxuan Yang [1] ✉, Bibek Bhujel [1], Daniel G. Chica[2], Evan J. Telford[2,3], Xavier Roy [2], Fatima Ibrahim [4], Mairbek Chshiev [4,5], Maxen Cosset-Chéneau [1] ✉ & Bart J. van Wees[1]

The magnetic proximity effect can induce a spin dependent exchange shift in the band structure of graphene. This produces a magnetization and a spin polarization of the electron/hole carriers in this material, paving the way for its use as an active component in spintronics devices. The electrostatic control of this spin polarization in graphene has however never been demonstrated so far. We show that interfacing graphene with the van der Waals antiferromagnet CrSBr results in an unconventional manifestation of the quantum Hall effect, which can be attributed to the presence of counterflowing spin-polarized edge channels originating from the spin-dependent exchange shift in graphene. We extract an exchange shift ranging from 27 – 32 meV, and show that it also produces an electrostatically tunable spin polarization of the electron/hole carriers in graphene ranging from − 50% to + 69% in the absence of a magnetic field. This proof of principle provides a starting point for the use of graphene as an electrostatically tunable source of spin current and could allow this system to generate a large magnetoresistance in gate tunable spin valve devices.

The 3d transition metal ferromagnets possess a large exchange interaction of the order of 1 eV, which has the effect of shifting the energy of their 3d electronic band structure depending on its spin[1]. This exchange shift leads to the presence of a magnetization, and of a spin polarization of the density of states at the Fermi level, which induces a spin polarization of the conductivity[2]. These have led to large magnetoresistance effects in spin valve structures[3] which exploit the spin polarization of the conductivity through the giant magnetoresistance (GMR), as well as the tunnel magnetoresistance (TMR)[4] which relies on the spin polarization of the density of states. GMR and TMR allowed the development of spintronics devices where information is stored in the parallel/antiparallel orientation of the magnetic layers, while these magnetoresistance effects provide effective means to readout the magnetic configuration. Writing the magnetic information however

remains challenging, as it relies on energy consuming processes, either using an external magnetic fields, spin transfer or spin orbit torques involving energy dissipative currents, to reverse the magnetization direction[5].

The possibility to electrostatically reverse the spin polarization would lift this limitation of spintronic devices[6]. This electrostatic tunability of the magnetization direction is however not possible in conventional metallic ferromagnets owing to their large carrier density. In contrast, graphene is an air stable metallic material, with a low carrier concentration which allows the electrostatic control of its transport properties[7]. In addition, the possibility to modify its band structure through proximity effects[8], and its spin filtering abilities[9], has led graphene to find its way into van der Waals heterojunctions[10,11]. However, owing to its lack of magnetic properties, graphene is not able to

[1]Zernike Institute for Advanced Materials, University of Groningen, 9747 AG Groningen, The Netherlands. [2]Department of Chemistry, Columbia University, New York, NY 10027, USA. [3]Department of Physics, Columbia University, New York, NY 10027, USA. [4]Univ. Grenoble Alpes, CEA, CNRS, Spintec, Grenoble 38000, France. [5]Institut Universitaire de France (IUF), Paris 75231, France. ✉e-mail: boxuan.yang@rug.nl; m.n.c.g.cosset-cheneau@rug.nl

detect or generate a spin current by itself and remains a passive component of spintronic devices.

The magnetic proximity effect (MPE)[12,13] has changed this paradigm. It has indeed been predicted[14,15] and demonstrated[12,16–23] that having graphene in contact with a magnetic material can induce an exchange energy shift in its band structure. It has been experimentally shown[24] that this shift leads to the appearance of a large spin polarization in graphene. This type of system could therefore be used in the predicted spin valves devices[25–28].

A spin polarization with a value of 14% has been reported in graphene interfaced with CrSBr[29], an air stable and insulating layered type A antiferromagnetic van der Waals material[30–32]. It is expected that a proper electrostatic adjustment of the Fermi energy could lead to an electrostatically sign reversible ± 100% spin polarization in this system owing to the electron/hole symmetry of its band structure. Such an effect has however not been reported so far. Indeed, while large exchange shifts have been reported in proximity-magnetized graphene[33], the region close to the charge neutrality Dirac point has not been explored. This is important since, as we will show, an efficient electrostatic control and sign reversal of the spin polarization can only be achieved when the Fermi energy lies in between the exchange shifted Dirac points.

In this paper we provide a direct demonstration of the tunability of the spin polarization of the carrier density, which reaches values of + 69% to − 50%. We use high field magnetotransport measurements to extract the exchange energy shift of the graphene band structure. The presence of the exchange shift modifies the well established energy spectrum of the graphene Landau Levels (LLs)[34], resulting in an unusual manifestation of the electronic transport in the Quantum Hall Effect (QHE) regime due to spin polarized counterflowing electron and hole edge channels. These high field quantum transport measurements allow us to obtain the specific values of the exchange shift using a self-consistent model of the spin polarized QHE edge channels, and of the spin polarization at zero magnetic field in proximitized graphene without relying on ferromagnetic contacts. Our results confirm

the possibility to electrostatically control the sign and amplitude of the spin polarization in graphene.

The device discussed in this work is shown in Fig. 1(a). It consists of a Hall bar-shaped bilayer graphene exfoliated on SiO₂, and on top of which a 100-nm thick CrSBr flake has been deposited [Fig. 1(b)]. The CrSBr consists of antiferromagnetically coupled layers, such that the layer in contact with graphene induces an uniform MPE. Magnetotransport measurement were performed at a temperature of 20 K and under the application of an out of plane magnetic field **B**. Additional magnetotransport measurements were performed at 180 K, above the Néel temperature of CrSBr. Details on the sample preparation and magnetotransport measurement procedure can be found in the "Method" section and in the Supplementary Information.

## Results and Discussion
### Theory
We performed four probe measurements of the magnetic field and gate dependence of longitudinal [Fig. 2(a)] and transverse (Hall) resistances [Fig. 2(b) and (c)]. The MPE in graphene is expected to lift the spin degeneracy[17,18,20,22,29], causing a modification of the conventional QHE measured in pristine graphene[35]. In this section we provide a theoretical framework which describes the effect of the exchange energy shift on the high field magnetotransport in bilayer graphene.

Figure 3a qualitatively shows the effect of the MPE on the graphene density of states. It induces an exchange energy shift $\pm \frac{\Delta}{2}$ of the electronic states depending on their spin. The electron and hole carrier densities can be indexed in separate populations according to their spin state. In the following they will be denoted $n_\sigma$ and $p_\sigma$ for electrons and holes with spin $\sigma = \uparrow$ or $\downarrow$. The density of these spin-polarized carriers depends on the position of the Fermi energy ($E_F$). When $E_F$ lies in between the two exchange shifted Dirac points, the system is in a two carrier regime with $n_\uparrow = 0$ and $p_\downarrow = 0$, so that there is a one-to-one correspondence between the electron (hole) states and the spin down (up) states. We call the position for which $n_\downarrow = p_\uparrow$ the charge compensation point (CCP), which corresponds to $E_F = 0$ meV.

The application of a strong magnetic field causes the formation of coexisting spin-polarized electron and hole LLs [Fig. 3(a)]. As depicted in Fig. 3(b), this results in the formation of edge channels with opposite flow direction owing to their different spatial dispersion relations close to the edges. These counterflowing edge channels will be populated up to the electrochemical potentials of the contacts from which they originate, and equilibrate in the voltage contacts[36,37]. In contrast to the case of the conventional QHE, the electrochemical potential of the side contacts will therefore not be a direct copy of the source or drain contacts. This will result in a non-zero longitudinal resistance and in a non-conventional transverse resistance as observed in Fig. 2.

We made a self-consistent model to describe the electronic transport (Supplementary Section V). The carrier density of the thermally-broadened exchange shifted LLs is first self-consistently evaluated as a function of the gate voltage. We thus obtain the Fermi energy as a function of the gate voltage and magnetic field. It is then possible to evaluate the finite temperature conductances $G_{N(P)} = (2e^2/h) N(P)$ of the spin-polarized electron ($N$) and holes ($P$) counter flowing edge channels between adjacent contacts. $N$ and $P$ correspond to the (thermal) occupation of the spin-polarized edge channels. We applied a gate voltage ranging from 0 V–7 V, for which the system was found to be in the spin-polarized two carrier regime. Finally, the voltage $V_k$ at the contact numbered $k$ is evaluated in the framework of the Landauer-Buttiker formalism[38]. The net charge current $I_k$ flowing in or out of the contact $k$ is:

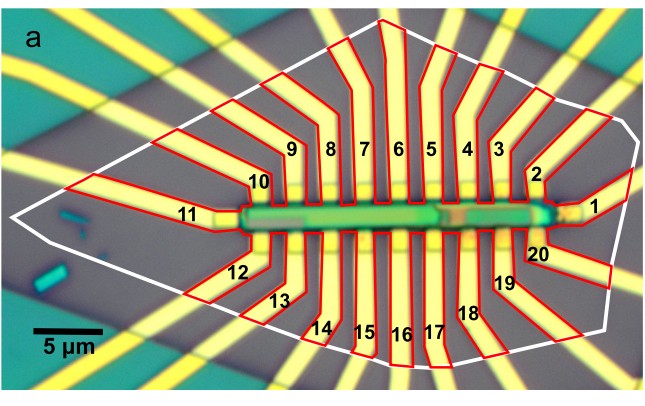

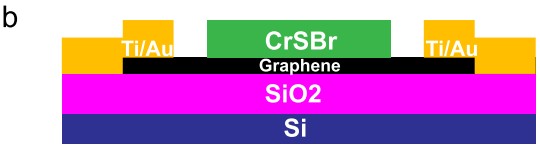

**Fig. 1 | Device structure. a** Optical micrograph of the bilayer graphene-CrSBr device. The white outline is the area covered by the exfoliated graphene flake, the red outline is the graphene Hall bar geometry after excess graphene was etched away. The green-yellow bar in the centre is the CrSBr flake, its non-uniform color is due to the irregular height of its surface. The large number of contacts allows to perform magnetotransport measurements on various electrically independent areas of the device to confirm the reproducibility of the magnetic proximity effect. **b** Cross section of the device presented in (**a**) showing the doped Si backgate separated from the device by an insulating SiO₂ layer.

$$I_k = \frac{2e^2}{h}\left[-(N+P)V_k + NV_{k+1} + PV_{k-1}\right] \qquad (1)$$

so that the voltage at contacts in which no net charge current flows ($I_k = 0$) will be a weighted average of its nearest neighbours:

$$V_k = \frac{NV_{k+1} + PV_{k-1}}{N+P} \qquad (2)$$

This model relies on the following assumptions: (1) The counterflowing edge channels with opposite spins do not equilibrate between the contacts. (2) There is no bulk transport pathway which allows current flow other than through the edge channels. (3) The counter-flowing edge channels are fully absorbed and the chemical potentials carried by them are equilibrated at the voltage contacts[36,39].

We now apply our model to the example diagram with six contacts shown in Fig. 3(c), in which the LLs and Fermi energies are given in Fig. 3(a), with their spatial dispersion relation in Fig. 3(b) for an illustrative set of parameters. The longitudinal resistance is (at $T = 0$ K) is:

$$R_{32} = \frac{V_3 - V_2}{I} = \frac{h}{2e^2}\frac{14}{63} \qquad (3)$$

and the transverse resistance:

$$R_{26} = \frac{V_2 - V_6}{I} = \frac{h}{2e^2}\frac{1}{3} \qquad (4)$$

Eq. (3) shows that in contrast to the conventional QHE, the longitudinal resistance is non zero. Eq. (4) predicts the appearance of transverse resistance plateau-like features at values which are not expected for the conventional QHE in non-magnetic graphene[40,41]. These features originate from the specific ways in which the top and bottom edge channel electrochemical potentials change when moving between contacts from the current source to the drain, and this is directly linked to the presence of counter propagating edge channels. This spatial dependence of the contact electrochemical potential for the device presented in Fig. 1(a) is reported in Supplementary Section V. An important consequence of this model is that for $N = P$ the electrochemical potential difference between the top and bottom contacts vanishes, leading to a zero transverse voltage. Furthermore the number of occupied electron/hole edge channels depends crucially on $\Delta$, which is the relevant parameter we will extract from the model.

**Comparison with measurement results**
In this section we discuss the results of the magnetotransport measurements, and their interpretation in the framework of the model presented in the Theory section. The bias current flows between contact 11 and 1 [see Fig. 1], while the longitudinal sheet resistance $R_{xx}$ was measured between contacts 9 and 6 [Fig. 2(a)]. The transverse resistance $R_{xy}$ was measured between two pairs of contacts: 10 and 12 (dataset A) [Fig. 2b], 8 and 14 (dataset B) [Fig. 2c].

We first focus on the low field results of dataset A. Shifting $V_g$ from 0 – 7 V changes the low field slope of $R_{xy}$ from negative to positive, with a sign change at 3.1 V, which is defined as $V_{g0}$, corresponding to the CCP. For $V_g < V_{g0}$, the majority of the carriers are spin-up holes. For $V_g > V_{g0}$ the majority of the carriers are spin-down electrons. A similar behavior was observed in dataset B where $V_{g0} \approx 3.3$ V. The low field value of $R_{xx}$ is weakly gate voltage dependent. This feature is interpreted in the framework of our model as a consequence of the exchange shift, which creates a gate voltage-independent total carrier density close to the charge compensation point (see below).

In contrast to the usual longitudinal resistance in bilayer graphene subjected to a large magnetic field, $R_{xx}$ does not go to zero [Fig. 2(a)]. It instead displays a minimum and then a sharp upturn (at 4.5 T for $V_g = 0$). The minimum progressively becomes less pronounced when increasing $V_g$ and the magnetic field at which it occurs decreases down to 3.3 T at $V_g = V_{g0}$, then increases again when $V_g > V_{g0}$. At a given gate

voltage, this minimum of $R_{xx}$ corresponds to an extremum of $R_{xy}$ which displays a similar voltage dependence [Fig. 2b, c]. The gate dependence of the magnetic field at which this feature occurs is reported in the Supplementary Section V. This behavior has not been observed in the high field magnetotransport of pristine graphene[35]. In the next section, the observed features are interpreted in the framework of the counterpropagating spin up and spin down channels model described in the Theory Section.

The carrier mobility $\mu$ was extracted from the low field longitudinal resistance [Fig. 2a] using $1/\rho = (n+p)e\mu$ with $\rho$ the resistivity, $-e$ the charge of the electron. Finally, $n = n_\uparrow + n_\downarrow$ and $p = p_\uparrow + p_\downarrow$ are the total electron and hole densities, the determination of which as a function of the applied gate is described in the Supplementary Section II. The mobility depends weakly on the gate voltage, with values ranging from $1.42 - 1.72 \, \text{m}^2 \cdot \text{V}^{-1} \cdot \text{s}^{-1}$ (see Supplementary Section III for the gate dependence).

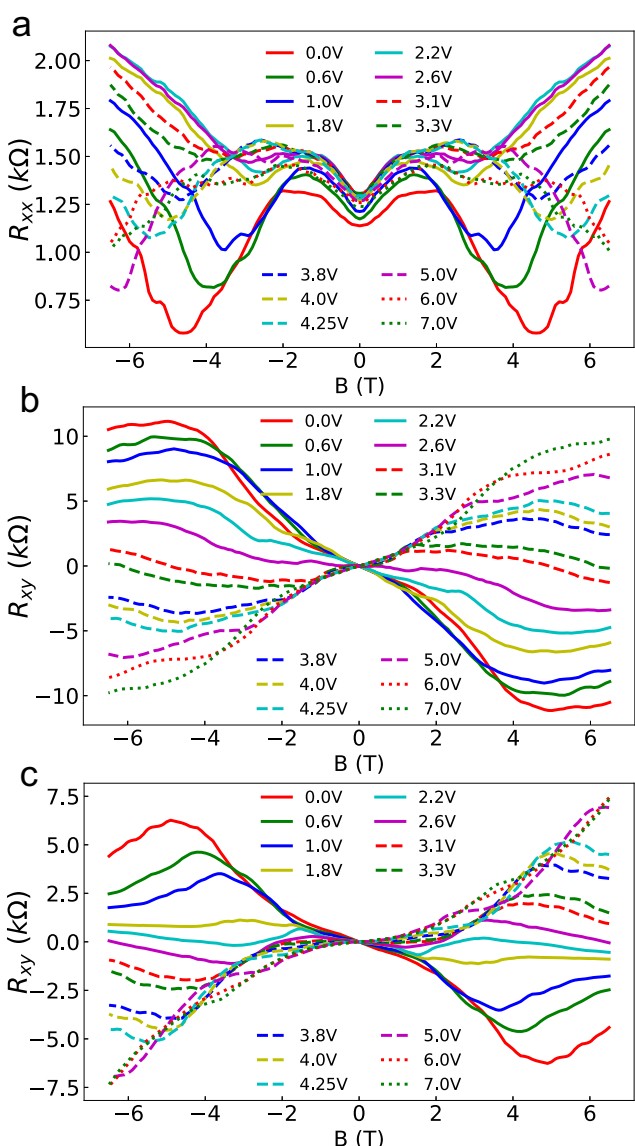

**Fig. 2 | Magnetotransport measurements. a** $R_{xx}$ measured between contacts 9 and 6 as a function of the gate voltage. This quantity is a square sheet resistance. **b** $R_{xy}$ measured between contacts 10 and 12 (dataset A). **c** $R_{xy}$ measured between contacts 8 and 14 (dataset B). $R_{xx}$ and $R_{xy}$ have been symmetrized and anti-symmetrized, respectively, with respect to the magnetic field to remove a small cross contamination between $R_{xx}$ and $R_{xy}$ due to a possible misalignment of the voltage contacts (see Supplementary Section VII for the raw data).

The main feature to be analyzed using our model is the non-monotonic behavior of the transverse resistance and its dependence on the gate voltage. In the following we illustrate the effect of the exchange energy shift on the magnetotransport in two carrier regime ranging from electron ($V_g > V_{g0}$) to hole ($V_g < V_{g0}$) majority. We then show that the observed features are well reproduced by our model calculations. An analysis similar to the one developed below is given for other sets of data in Supplementary Section VI.

In Fig. 4a, b, we show the longitudinal resistance from Fig. 2a and transverse resistance extracted from the dataset A [Fig. 2b] at $V_g = 5$ V. The system is in a spin down-electron majority regime ($n_\downarrow > p_\uparrow$) with a carrier density estimated in the Supplementary Section II. We used the model described in the Theory Section to calculate the transverse resistance, with the energy shift $\Delta$ as a fitting parameter. The calculated $\Delta$ value which gives a $R_{xy}$ best reproducing the measured one displayed in Fig. 4b is 27 meV. In Fig. 4c, d, we show the LL positions calculated using this parameter at $B = 2.8$ and 6.2 T, respectively. We also show the position of the Fermi energy at $V_g = 5$ V. Fig. 4c shows the LL energies calculated with a magnetic field of 2.8 T. Here the only populated LLs

are the spin down $n = (0, 1)$ and $n = 2$, and the spin up $n = (0, 1)$ (here $n$ refers to the bilayer graphene LL number). When the magnetic field goes above 6.2 T, as shown in Fig. 4d, the spin down $n = 2$ LL has moved above the Fermi energy and became depopulated. The occupation of spin up ($P$) and down ($N$) channels therefore becomes approximately the same, causing $R_{xy}$ to decrease. Since the measurements have been carried out at 20 K, the spin up $n = 2$ channel is still partially populated for a magnetic field of 6.5 T, so that a complete cancellation of $R_{xy}$ is not reached. The longitudinal resistance in Fig. 4(a) could not be reproduced using our model owing to the possible presence of bulk transport paths. We show in the Supplementary Section VI that our model nevertheless qualitatively predicts the upturn of $R_{xx}$ at 6.2 T.

The longitudinal and transverse resistance from dataset B [Fig. 2c] at $V_g = 1.8$ V, close to the charge compensation point ($V_{g0} = 2.2$ V) in this region of the sample, are shown in Fig. 4e, f, respectively. At this gate voltage there is a spin up-hole majority ($p_\uparrow > n_\downarrow$). We used the procedure described above and find $\Delta = 32$ meV. Here, although our model qualitatively reproduces the features of $R_{xy}$, the calculated amplitude is much larger than the measured one. We believe that this discrepancy

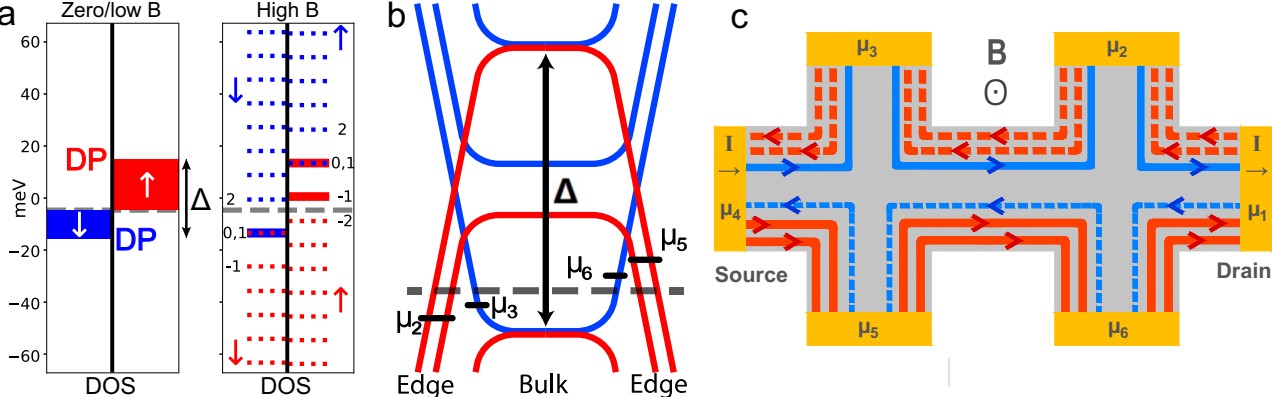

**Fig. 3 | Exchange shifted densities of state, Landau level energies and counterflowing edge channels in proximity-magnetized graphene. a** Left: Spin-up polarized holes (red) and spin-down polarized electrons (blue) densities of state of proximity-magnetized bilayer graphene at B = 0 T and Δ = 27 meV. The exchange shifted Dirac points are marked as DP. $E_F$ (grey dashed line) is located in the 2 carrier regime, corresponding to $V_g - V_{g0} = -1.3$ V. Right: LL energies in bilayer graphene at $B = 2.2$ T with Δ = 27 meV. Blue and red lines corresponds to electron and hole LLs, respectively. Solid (dashed) lines are occupied (unoccupied) energy levels. The arrows indicate the spin state. The numbers indicate the indices of the spin polarized LLs, with the energy of the degenerate $n = (0, 1)$ LLs at the exchange shifted Dirac points and independent on the magnetic field. **b** LL spatial dispersion and edge state chemical potentials of the sample section located between contacts 3 and 2, and 5 and 6. The Fermi energy is the same as in (**a**). **c** Coexisting spin-down electron (blue) and spin-up hole (red) quantum Hall edge channels. Channels originating from the source contact (4) are in solid lines, while those originating from the drain (1) are in dashed lines. The $\mu_i$ with $i = 1$ to 6 are the contact electrochemical potentials.

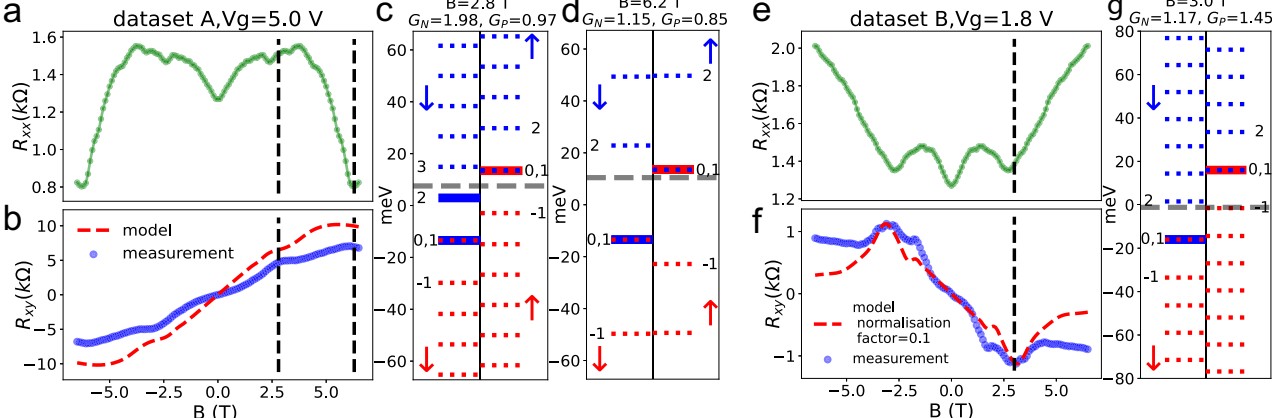

**Fig. 4 | Comparison of the model prediction and the transport measurements. a** Sheet resistance $R_{xx}$ and **b** $R_{xy}$ from dataset A. The gate voltage is $V_g = 5$ V. The dashed line corresponds to the $R_{xx}$ calculated using Δ = 27 meV. **c** LLs energies calculated using Δ = 27 meV and $B = 2.8$ T, corresponding to the first vertical dashed line on the left. **d** same as **c** with $B = 6.2$ T, corresponding to the second vertical dashed line on the left. **e** $R_{xx}$ and **f** $R_{xy}$ from dataset B, $V_g = 1.8$ V. The dashed line corresponds to the transverse resistance using Δ = 32 meV. **g** LLs energies calculated using Δ = 32 mev and $B = 3$ T.

originates from our simplifying assumption of edge channel-only electronic transport. The calculated results are therefore normalized for the experimental maximum in order to qualitatively compare the main features of $R_{xy}$. Both $R_{xx}$ and $R_{xy}$ display an extremum at $B \approx 3$ T. The corresponding Fermi and LLs energies are shown in Fig. 4g. At this field, the $n = -1$ spin up and $n = 2$ spin down LLs have moved respectively below/above the Fermi energy, such that only the $n = (0, 1)$ LLs for these spin states remain occupied, and $N$ and $P$ both come close to 1. This same number for the electron and hole edge channels causes the transverse resistance to further decrease when the magnetic field goes above 6.5 T. Another noteworthy feature in Fig. 4(f) is the small shoulder at $B = 1.7$ T, this is caused by the $n = -2$ spin up channel moving below $E_F$ at this field value. This feature is also reproduced by our model. When moving the Fermi level from the electron to the hole-majority regime, we observed that the features described in Fig. 4 are anti-symmetric in terms of relative gate voltage $Vg - Vg_0$ for the same magnetic fields (Supplementary Section VI). This indicates that the sign and magnitude of the exchange shift is independent on the Fermi energy in the range of gate voltages used in this study.

We observed a variation of $\Delta$ extracted from different areas of the device, ranging from $27 - 32$ meV. This indicates that the MPE is not fully homogeneous across our device. These values are however in agreement with the 20 meV extracted previously on a similar system using a different method[29]. The energy shift corresponds to an effective magnetic field $B_{eff}$ with $\Delta = g\mu_B B_{eff}$[22], in which $g \approx 2$ is the Landé factor and $\mu_B$ the Bohr magneton. The resulting effective fields vary from 231 to 276 T. During the measurements our applied magnetic field never exceeds 7 T, such that the Zeeman and valley splittings can be neglected[42,43].

The exchange energy shift originates from the MPE and is therefore also present at low and zero magnetic field. Figure 5a shows the calculated electron/hole density as a function of the gate voltage. The presence of an exchange shift implies the existence of a two carrier regime for $|V_g - V_{g0}| < 4.5$ V.

The Hall coefficient $R_H$ for two carrier $n$ and $p$ with the same mobilities is expressed as $eR_H = (n - p)/(n + p)^2$ (see Supplementary Section II) and corresponds to the slope of the transverse resistance measured at low field. The Hall coefficient is expected to display a smooth crossing and a sign change at the CCP when an exchange energy shift is present in the graphene band structure. It then reaches a maximum and decreases when the system enters the single carrier regimes ($n = 0$ or $p = 0$). In Fig. 5b we plot the Hall coefficient calculated using the $n$ and $p$ extracted from the model described in the Supplementary Section II, and using $\Delta = 27$ meV, extracted from the high field magnetotransport measurements. Figure 5b also shows the measured low field Hall coefficient. We find a good agreement between our model and the experimental values close to the charge compensation point, while a deviation occurs away from the CCP.

The exchange shift induces an equilibrium magnetization in graphene, expressed as $M_{Gr} = g\mu_B(n_\uparrow - p_\uparrow + p_\downarrow - n_\downarrow)$. In bilayer graphene, $M_{Gr} = g\mu_B(m_{eff}/\pi\hbar^2)\Delta$, which is independent of the Fermi energy since the density of states is approximately energy independent in this system. We however expect this magnetization to be energy dependent in monolayer graphene. This offers the possibility for the electrostatic manipulation of the magnetization. Another relevant quantity is the spin polarization, which is defined as:

$$P = \frac{n_\uparrow + p_\uparrow - n_\downarrow - p_\downarrow}{n_\uparrow + p_\uparrow + n_\downarrow + p_\downarrow}(\times 100\%) \qquad (5)$$

In the following we use the extracted $\Delta = 27$ meV to evaluate $P$.

In the two carrier regime, there is a one to one correspondence between the electron (hole) and spin down (up) densities, such that $P = (p_\uparrow - n_\downarrow)/(p_\uparrow + n_\downarrow)$. Here, $V_g < V_{g0}$ corresponds to a hole-spin up majority regime, which translates into a positive spin polarisation [Fig. 5c]. Similarly, $V_g > V_{g0}$ corresponds to a negative spin polarisation at the Fermi energy. At $T = 0$ K, the spin polarization can reach $\pm 100\%$ at the spin down (up) Dirac point since $n_\downarrow = 0$ ($p_\uparrow = 0$). The polarization is zero at the charge compensation point where $n_\downarrow = p_\uparrow$, i.e. for $V_g = V_{g0}$. When moving outside the two carrier regime for $|V_g - V_{g0}| > 4.5$ V, the spin up and down density of both electrons and holes increase, causing a decrease of the spin polarization at large positive and negative gate voltage [Fig. 5c]. In the two carrier regime, the Hall coefficient writes $eR_H = -P/(n + p)$, hence providing a direct measurement of the spin polarization. In Fig. 5c we show the spin polarization evaluated using the measured Hall coefficient and the calculated $n$ and $p$. We observe a polarization ranging from $+69\%$ at $V_g = 0$ V to $-50\%$ $V_g = 7$ V. At zero magnetic field the CrSBr magnetization is transverse to the flake long axis in Fig. 1a, and sets the direction of the graphene spin polarization[29].

The conductivity of spin $\uparrow$ ($\downarrow$) carriers is expressed as $\sigma_{\uparrow(\downarrow)} = e(n_{\uparrow(\downarrow)} + p_{\uparrow(\downarrow)})\mu_{\uparrow(\downarrow)}$. The symmetry of the quantum Hall magnetoresistance when moving from the electron to hole-dominated regime indicates that the exchange shift modifies the electron and hole LL energies in the same manner. We therefore expect the mobility to be independent of the spin of the carrier, such that $\mu_\uparrow = \mu_\downarrow$. In this case, the spin polarization of the carrier density and the spin polarization of the conductivity are directly related.

It is now possible to evaluate the GMR of a lateral all-graphene electrostatically-modulated heterostructure. We use a spin polarization of $\pm 60\%$, leading to a gate-tunable GMR of up 50% for systems with lateral lengths comparable to the spin diffusion length. For a system with an energy-dependent density of states, such as monolayer graphene, it is also possible to view the polarization in our system as a spin polarization of the density of states at the Fermi energy. In this case, one may expect a tunneling magnetoresistance (TMR)[44] of 112%. Here we propose as a proof of concept the magnetic tunnel junction made from proximity-magnetized graphene shown in Fig. 6, in which the switching between "parallel" and "anti-parallel" states can be achieved by applying different gate voltages to the graphene layers.

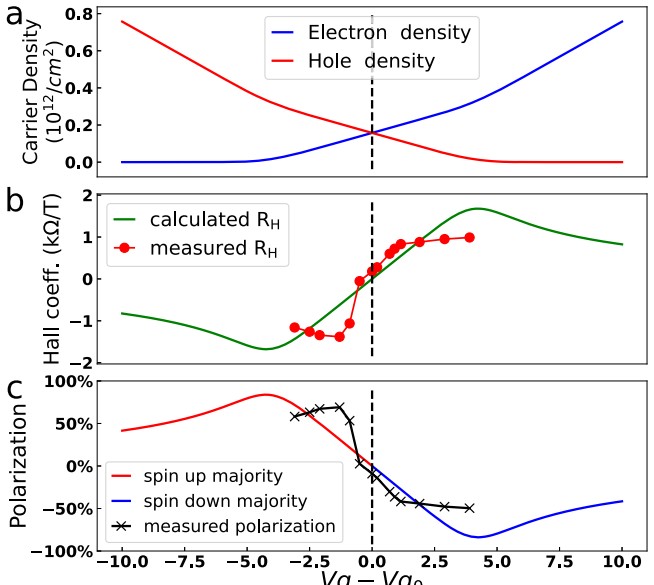

**Fig. 5 | Extraction of the spin polarization of the carrier density as a function of the gate voltage. a** Calculated electron and hole densities. **b** Calculated (dashed line) and the measured (red dots) Hall coefficient $R_H$ from dataset A as a function of the gate voltage, and **c** calculated (colored line) and measured (black line) spin polarization. A temperature of 20 K and an exchange shift $\Delta = 27$ meV were used for the calculations.

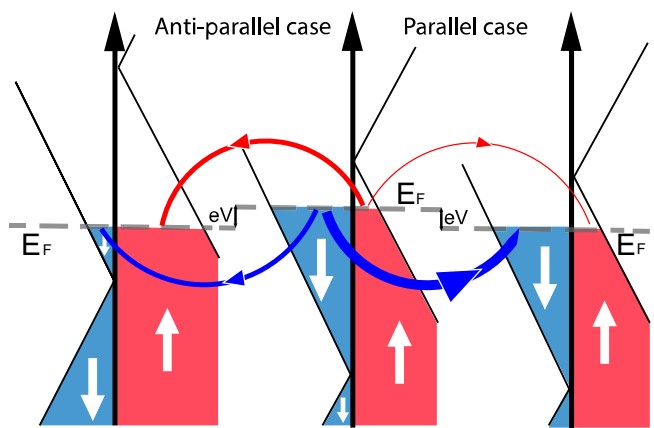

**Fig. 6 | Proposed electrostatically-controlled tunnel junction made of proximity-magnetized graphene.** The high resistance ("anti-parallel") case is on the left while the low resistance ("parallel") case is on the right. The thickness of the arrows represent the magnitude of the spin up (red) and down (blue) tunneling current. An electrostatic bias $eV$ is applied between the graphene layers to achieve the high and low resistance states.

We performed low and high field magnetotransport measurement in proximity-magnetized graphene, and extracted an exchange shift of 27 – 32 meV, and showed that it results in a spin polarization of the carrier density. The spin polarization is tunable electrostatically, and reached values between +69 and −50%.

Although our modelling matched closely, both qualitatively and quantitatively, with our magnetotransport measurements, discrepancies between the model and the experiment remain. The most striking one is the deviation of the measured Hall coefficient from its predicted values when moving away from the charge neutrality point. Our model assumes that the only effect of the MPE on the band structure of graphene is to cause a constant exchange energy shift, which does not depend on the carrier type or on the electronic state energy. This assumption might be too restrictive to fully account for the details of our data in view of theoretical works predicting a carrier dependent exchange shift, as well as a possible gap opening at the graphene Dirac point[45]. Furthermore, an energy dependence of the exchange shift can be expected in this type of systems[46,47], and the MPE strongly depends on the stacking between graphene and CrSBr as shown in Supplementary Section II.D. Further refinement of our model is possible to test these theoretical predictions.

Overall, the proximity-magnetized graphene system appears in this work as promising for the development of spintronic devices based on the electrostatic tunability of magnetic properties. It indeed displays possibilities for the realization of gate-controllable spin valves and spin filters, and will be of great significance in the exploitation of spin-charge interconversion phenomena in van der Waals heterostructures[8,48,49] owing to the optimized spin injection efficiency offered by the combination of a large electrostatically-tunable spin polarization with the long spin relaxation length of graphene[29]. We also predict that proximitized graphene hosts an equilibrium magnetization, which could be modified upon the application of an a.c. or d.c. gate voltage. We show that our results can be used to electrostatically generate the GMR and TMR effects.

## Methods

The device presented in Fig. 1 was fabricated dry-transferring a CrSBr flake (thickness ~ 100 nm) on top of a bilayer graphene flake which was exfoliated on a $Si/SiO_2$(285 nm) substrate. Ti/Au contacts were deposited on the graphene in a Hall bar pattern, and the excess graphene outside of the Hall bar was etched away. We used conventional lock-in measurement techniques, with a bias current of 10 – 100 nA and frequency 13.7 Hz. A gate voltage $V_g$ ranging from 0 to 7 V was applied between the $p^{++}$ doped Si substrate and graphene during the measurements.

## Data availability

All data supporting this study are available at https://doi.org/10.5281/zenodo.10654565.

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

## Acknowledgements

We acknowledge the financial support of the Zernike Institute for Adv. Mater. and the European Union's Horizon 2020 research and innovation program under Grant Agreement No. 785219 and No. 881603 (Graphene Flagship Core 2 and Core 3). This project is also financed by the NWO Spinoza prize awarded to B.J. van Wees. by the NWO and has received funding from the European Research Council (ERC) under the European Union's 2DMAGSPIN (Grant agreement No. 101053054). Synthesis of the crystals was supported by the National Science Foundation (NSF) through the Columbia University Materials Research Science and Engineering Center (MRSEC) on Precision-Assembled Quantum Materials DMR-2011738. Electrical transport measurements at Columbia were supported as part of Programmable Quantum Materials, an Energy Frontier Research Center funded by the US Department of Energy, Office of Science, Basic Energy Sciences, under award DE-SC0019443.

## Author contributions

B.J.v.W. and B.Y. conceived the experiment. B.Y and B.B. fabricated the device, performed the measurements and developed the model. B.J.v.W., B.Y., B.B. and M.C.C. analyzed the data and discussed the experimental results. B.J.v.W., B.Y. and M.C.C. wrote the manuscript. D.G.C., E.J.T. and X.R. performed the growth of the CrSBr crystals. All authors discussed the results and commented on the manuscript. F.I. and M.C. performed the DFT calculations.

## Competing interests

The authors declare no competing interests.
