## [Peer Review File · Nature Communications]

Electrostatically controlled spin polarization in Graphene-CrSBr magnetic proximity heterostructuresREVIEWER COMMENTS

Reviewer #1 (Remarks to the Author):

This work reports the electronic transport properties in Graphene/CrSBr heterostructure manipulated by electrostatic gating. Especially, the tunable spin-polarization by gating voltage has been demonstrated. Overall, this manuscript is interesting and the developed theoretical model seems consistent with experimental results. It can be further considered for publication if the following concerns can be addressed:

- 1.If it is possible to perform DFT calculation to extract proximity exchange energy and the gate controlled sign change of spin-polarization CrSBr/Graphene heterostructure to offer more intensive comparison with experiments?
- 2.The spin-polarization of carrier in Graphene has been measured by Hall effect in this work. If it is also possible to directly provide longitudinal transport evidence?
- 3.The designed tunnel junction shown in Fig.6 seems impractical because the spin-polarization of carrier in Graphene can only be changed by applying gate voltage, which it's volatile. This design can not compete with the magnetic tunnel junctions like CoFeB/MgO/CoFeB, which is nonvolatile. Instead, if it is possible to design some kind of spin-transistor based on the gate voltage controlled spin-polarization in Graphene?

Reviewer #2 (Remarks to the Author):

In this paper, authors show a spin polarization of the electron/hole carriers in bilayer graphene interfacing with the van der Waals antiferromagnet CrSBr by the magnetic proximity effect. More importantly, the electrostatic control of this spin polarization in graphene has been demonstrated, paving the way for its use as an active component in spintronics devices. They show the unconventional manifestation of the quantum Hall effect, which can be attributed to the presence of counterflowing spin-polarized edge channels originating from the spin-dependent exchange shift in graphene. This provides a starting point for the use of graphene as an electrostatically tunable source of spin current and could allow this system to generate a large magnetoresistance in gate tunable spin valve devices. However, some key issues need to be clarified before making a decision.

- 1.The magnetic proximity effect essentially arises from the orbital interaction between spin polarized Cr atoms in CrSBr and graphene carbon atoms. Due to the van der Waals contact between graphene and CrSBr, the magnetic proximity effect may not be strong enough to cause large spin polarization in graphene. A spin polarization only 14% has been reported in graphene interfaced with CrSBr in previous experiment. In this experiment this value is up to 69%. This has led to doubts about the experimental conclusions. I suggest the author supplement the results of theoretical calculations, such as calculating the band spin splitting of graphene in the heterostructure composed of graphene and CrSBr through first principles calculations. This can show band spin splitting and Fermi level, explaining these results.
- 2.The author's experiment shows that an external electrostatic field can change the spin polarization sign. This change is related to the Fermi level and spin splitting band structure. These experimental results can be understood through theoretical calculations.
- 3.Has the author characterized the magnetic easy axis of CrSBr in experiment? Previous experiments have shown that the magnetic easy axis is in-plane and can be controlled by a magnetic field. Magnetic proximity effect can be effected by magnetic easy axis because it

can change magnetic exchange interaction. Does the direction of the magnetic easy axis affect the experimental results and conclusions?

Reviewer #3 (Remarks to the Author):

Report for "Electrostatically controlled spin polarization in Graphene-CrSBr magnetic proximity heterostructures" by Boxuan Yang, et al.

The authors proposed that the magnetic proximity effect induces a spin-dependent exchange shift in the band structure of graphene, resulting in magnetization and spin polarization of electron/hole carriers. Consequently, electrostatic manipulation can be employed to exert control over the spin polarization in graphene. To validate this conclusion, van der Waals antiferromagnet CrSBr-graphene heterostructures were fabricated by the authors, and magnetotransport measurements were conducted at both low and high magnetic fields. An exchange shift ranging from 27 to 32 meV was observed, demonstrating its correlation with the spin polarization of carrier density. Furthermore, they successfully demonstrated the electrostatic modulation of spin polarization, achieving a wide range of values from +69% to -50%. These results are very important to develop the spintronic devices based on the electrostatic tunability of magnetic properties. Therefore, this work is meaningful and should be published at Nature Communications. However, some issues should be addressed in an adequate manner before I can fully support the publication of this paper.

1. The cross-sectional view of CrSBr-graphene heterostructures has not been documented in the article. Could you kindly provide a detailed the cross-sectional image of heterostructures?
2. Whether the magnetic proximity effect is influenced by the packing direction and thickness of the two materials, and if so, please provide relevant evidence; If not, kindly explain the underlying rationale.
3. The authors only present the test results of a single device, which lacks universality. Could the authors conduct experiments on multiple devices to validate their proposed theoretical model?
4. The simulation results are in qualitative and quantitative agreement with their measurements of magnetic transport. However, there still exist disparities between the model and the experimental findings. Can the authors elaborate on the reasons for disparities?
5. The authors posit that the disparity between modeling and experimental results primarily stems from the deviation of the measured Hall coefficient from its predicted values as one moves away from the charge neutrality point. Their model assumes that the sole impact of MPE on graphene's band structure is a constant exchange energy shift, which remains independent of carrier type or electronic state energy. However, the author lacks comprehensive experimental data to substantiate the aforementioned conclusion. Can the author furnish it?

Reviewer #1 (Remarks to the Author):

This work reports the electronic transport properties in Graphene/CrSBr heterostructure manipulated by electrostatic gating. Especially, the tunable spin-polarization by gating voltage has been demonstrated. Overall, this manuscript is interesting and the developed theoretical model seems consistent with experimental results. It can be further considered for publication if the following concerns can be addressed:

We thank the Reviewer for his/her appreciation of the interest of our work, and address the concerns in the following comments.

1.If it is possible to perform DFT calculation to extract proximity exchange energy and the gate controlled sign change of spin-polarization CrSBr/Graphene heterostructure to offer more intensive comparison with experiments?

First, we would like to emphasize the physical picture used in our paper to analyze our magnetotransport measurements. The presence of an exchange shift induced by the magnetic proximity effect shifts the energy of the electronics states with opposite spin polarization. This also results in a shift of the Landau level energies depending on their spin.

Our analysis is based on the modification of the well calibrated Landau level energy spacing in pristine graphene by the magnetic proximity effect. This results in a change in the high field magnetotransport in proximitized graphene with respect to pristine graphene as detailed in the Supplementary Information (section V). From the comparison of these results with our model calculations we extracted the exchange shift in proximitized graphene. These results are independent of the details of the transport properties in proximitized graphene.

We then confirmed the spin polarization using low field magnetotransport measurements, which we compared to model calculations in the classical transport regime using the exchange shift extracted from the high field measurements. Since electrons and holes have opposite spin polarization in the two carrier regime, a modification of the majority carrier when crossing the charge compensation point results in a sign change of the graphene spin polarization. Once again, this gate control of the graphene spin polarization originates from very general considerations and do not necessitate to know the details if the proximitized graphene band structure.

Consequently, the extraction of the exchange shift and the gate voltage dependence of the polarization are a direct consequence of the physical picture used in our paper and in our opinion do not necessitate additional DFT computations.

Although DFT calculations of bilayer graphene proximitized with CrSBr are not strictly necessary to support our claim, we have been in touch with Mairbek Chshiev and Fatima Ibrahim (Université Grenoble Alpes and CEA Grenoble) regarding the possibility to perform said calculations. This work is currently in progress, and we will summarize here the preliminary results obtained for calculations done for a bilayer graphene layer in contact with a CrSBr bilayer.

1. The presence of the CrSBr layer is expected to induce a hole doping in graphene, which has been observed in our measurements. We however would like to emphasize that it could originate from extrinsic doping in the actual device used for this study.
2. The proximity effect induces an exchange shift which strongly depends on the van der Waals gap between graphene and CrSBr, with a modification of one order of magnitude when changing this gap by a few Angstroms.
3. The exchange shift appears to depend on the relative orientation and twisting between the graphene and CrSBr crystal axis.

We emphasize that these calculations have been performed for bilayer CrSBr on top of graphene. The CrSBr layer used in our experiments is much thicker (100 nm), such that the conclusion of these calculations should be considered with care. They however overall support our finding that the proximity effect between CrSBr and bilayer graphene can induce an exchange shift in the graphene band structure of the order of magnitude which we observed, and therefore can produce a spin polarization as discussed in our paper.

Because these calculations are still in progress, we do not report them in full, but instead summarized their preliminary conclusions in the Supplementary Information (Section II.D):

In the following we summarize the preliminary results of band structure calculations of bilayer graphene in contact with a bilayer of CrSBr obtained using density functional theory calculations. The calculations were first performed with CrSBr a-axis along the graphene armchair direction. In agreement with previous calculations [4], it was found that a parallel configuration of the two CrSBr layers magnetization is more stable than an antiparallel configuration. In addition, the presence of CrSBr is expected to results in a hole doping of graphene. The band structure of graphene displays a small exchange shift of 0.2 meV for a van der Waals gap $d=3.3$ Å. The exchange shift goes up to 1 meV for $d=2.5$ Å.

The effect of graphene stacking direction was then studied by performing the calculations with a CrSBr c-axis rotated by 30° with respect to the graphene armchair direction. An exchange shift of 7 meV is observed for $d=3.3$ Å and should increase when reducing the van der Waals gap.

The magnetic proximity effect induced by CrSBr on the graphene band structures therefore depends on the stacking direction and van der Waals gap between these two materials. In addition, the CrSBr layer used in the devices presented in Fig. 1 of the main text is thicker than the bilayer used in the calculations. Further work will be needed for a full theoretical understanding of the coupling between graphene and CrSBr.

We acknowledge the support from Mairbek Chshiev and Fatima Ibrahim in the acknowledgment section (line 472):

We thank Fatima Ibrahim and Mairbek Chshiev for useful discussions.

2.The spin-polarization of carrier in Graphene has been measured by Hall effect in this work. If it is also possible to directly provide longitudinal transport evidence?

A crucial aspect of this work is the identification of the exchange splitting and resulting spin polarization in proximitized graphene by using only charge transport measurements. By focusing on the Hall transport measurement, we make use of a specificity of the spin polarization in magnetic proximitized graphene in the gate voltage interval used for this study. In this gate voltage range, the carriers from the conduction (electrons) and valence (holes) bands, carry opposite spins. Since the sign of the transverse resistance depends on the type of the majority carrier, its sign and amplitude is therefore directly related to the spin polarization. Hall measurements therefore provide a direct access to the spin polarization in graphene by separating the contribution from carriers with opposite spin polarization. In contrast, low field longitudinal measurements do not allow to separate the contribution of the spin polarized electron and hole carriers, and do not give information regarding the spin polarization in graphene.

An identification of the spin polarization in graphene by longitudinal transport measurement could efficiently be done by realizing the spin valve structure proposed in Refs. 24-27 of the main text, were the carrier density can be controlled separately in different regions of the device. These experiments are currently underway, and are outside the scope of this paper.

3.The designed tunnel junction shown in Fig.6 seems impractical because the spin-polarization of carrier in Graphene can only be changed by applying gate voltage, which it's volatile. This design can not compete with the magnetic tunnel junctions like CoFeB/MgO/CoFeB, which is nonvolatile. Instead, if it is possible to design some kind of spin-transistor based on the gate voltage controlled spin-polarization in Graphene?

We would first like to emphasize the fact that the device proposed in Fig. 6 is a proof of concept of a possible proximitized graphene-based tunnel junction. We make this clear by adding the following remark in 422 to 426:

Here we propose as a proof of concept the magnetic tunnel junction made from proximity-magnetized graphene shown in Fig. 6, in which the switching between "parallel" and "anti-parallel" states can be achieved by applying different gate voltages to the graphene layers.

The realization of such a device, although challenging, is furthermore not impossible. There have been recently developments regarding the interplay between the ferroelectric and spin degrees of freedom [Noel et al, Nature 580, 483–486 (2020)]. In addition, novel ferroelectric van der Waals materials are currently being introduced. The possibility to use the polarization of such ferroelectric materials to control the Fermi energy would therefore open the way to realize an electrostatically-controlled non-volatile tunnel junction. We however believe that this last point is outside the scope of this paper, and should not be referred to.

Reviewer #2 (Remarks to the Author):

In this paper, authors show a spin polarization of the electron/hole carriers in bilayer graphene interfacing with the van der Waals antiferromagnet CrSBr by the magnetic proximity effect. More importantly, the electrostatic control of this spin polarization in graphene has been demonstrated, paving the way for its use as an active component in spintronics devices. They show the unconventional manifestation of the quantum Hall effect, which can be attributed to the presence of counterflowing spin-polarized edge channels originating from the spin-dependent exchange shift in graphene. This provides a starting point for the use of graphene as an electrostatically tunable source of spin current and could allow this system to generate a large magnetoresistance in gate tunable spin valve devices. However, some key issues need to be clarified before making a decision.

We thank the Reviewer for this clear summary of our manuscript, and we provide below the clarifications asked by the referee.

1. The magnetic proximity effect essentially arises from the orbital interaction between spin polarized Cr atoms in CrSBr and graphene carbon atoms. Due to the van der Waals contact between graphene and CrSBr, the magnetic proximity effect may not be strong enough to cause large spin polarization in graphene.

We do not fully understand what the Reviewer means when stating that “the magnetic proximity effect may not be strong enough to cause large spin polarization in graphene”. We actually show the opposite. A spin polarization ranging from -50% to +69% can be induced in graphene by the magnetic proximity effect.

In our paper, we extract a 30 meV exchange shift induced by the magnetic proximity effect in graphene. Although the exchange shift is much smaller than 1 eV reported in 3d ferromagnetic metals, it is enough to induce a large spin polarization in graphene. As explained in our manuscript, this results from the electron/hole semimetal character of graphene, which causes electron and hole type carriers to have opposite spins in the gate voltage range used by our study. Using a gate voltage which positions the Fermi energy such that the density of electrons or holes vanish results in a maximum spin polarization, which can reach $\pm 100\%$ at 0 K (and $\pm 70\%$ at 20 K) [Fig. 5]. These large spin polarization values can be reached as long as the thermal energy is smaller than exchange splitting.

A spin polarization only 14% has been reported in graphene interfaced with CrSBr in previous experiment. In this experiment this value is up to 69%. This has led to doubts about the experimental conclusions.

Once again, the problem raised by the Reviewer is not fully clear to us. The 14% reported in Ghiasi et al, [*Nat. Nanotechnol.* **16**, 788–794 (2021)] was achieved in a graphene-CSB device in which the Fermi energy could not be modified by a gate voltage. Consequently, and in contrast to the present paper, the Fermi energy was not tuned to reach the maximum spin polarization. This explains why a smaller spin polarization was extracted in the report by Ghiasi et al.

Therefore, the previous measurement of a 14% spin polarization in graphene-CSB does not disprove either the analysis of Ghiasi et al, or the one performed in the present paper. The fact

that both these studies extracted similar exchange shifts points in contrast toward the consistency of the analysis.

I suggest the author supplement the results of theoretical calculations, such as calculating the band spin splitting of graphene in the heterostructure composed of graphene and CrSBr through first principles calculations. This can show band spin splitting and Fermi level, explaining these results.

As argued in the response to the question 1 of Reviewer 1, our conclusion that an exchange shift is responsible for the spin polarization originates directly from the numerical calibration provided by the modification of the well-known Landau level spacing in graphene by an exchange shift. These conclusions do not necessitate support from DFT calculations.

As stated in the response to the question 1 of Reviewer 1, we however obtained preliminary DFT calculations on bilayer graphene in contact with bilayer CrSBr from Fatima Ibrahim and Mairbek Chshiev. These calculations support our findings.

2. The author's experiment shows that an external electrostatic field can change the spin polarization sign. This change is related to the Fermi level and spin splitting band structure. These experimental results can be understood through theoretical calculations.

We are not sure to understand what the Reviewer means by “These experimental results can be understood through theoretical calculations”. The possibility to change the sign and amplitude of the spin polarization is the main focus of our paper, and has been attributed to the exchange shift (or spin splitting) in the band structure of graphene caused by the magnetic proximity effect with CrSBr. The application of a gate voltage changes the Fermi energy and modifies the spin polarized electron and hole density, leading to a modification of the spin polarization as reported in Fig. 5 and emphasized in the answer to the first question of the Reviewer 1.

Our model, detailed in the Section II of the Supplementary information, explains these results in terms of exchange shift and Fermi energy, such that we do not believe that additional theoretical calculations are needed.

3. Has the author characterized the magnetic easy axis of CrSBr in experiment? Previous experiments have shown that the magnetic easy axis is in-plane and can be controlled by a magnetic field. Magnetic proximity effect can be affected by magnetic easy axis because it can change magnetic exchange interaction. Does the direction of the magnetic easy axis affect the experimental results and conclusions?

The easy magnetic axis of CrSBr sets the polarization direction of graphene. However, our analysis of the magnetotransport measurements does not depend on this direction.

The magnetic easy axis of CrSBr has been characterized in Ghiasi et al, Nat. Nanotechnol. 16, 788–794 (2021). At zero field, it is transverse to the current flow direction in our Hall bar geometry, and saturates out of plane at around 1.5 T during our magnetotransport measurements with an out of plane field. We make a note on the orientation of the spin polarization at low field in line 401 to 404:

At zero magnetic field the CrSBr magnetization is transverse to the flake long axis in Fig. 1, and sets the direction of the graphene spin polarization [28].

The specific value of the exchange shift could in principle depend on the CrSBr magnetization direction. This would result in a different exchange shift for an out of plane (large magnetic field) and in plane (low magnetic field) magnetization direction and could lead to the small discrepancy observed between the experiments and our calculations (see Fig. 5). We however do not expect this effect to be important, owing to the good agreement observed between our low field measurements and calculations.

Reviewer #3 (Remarks to the Author):

Report for "Electrostatically controlled spin polarization in Graphene-CrSBr magnetic proximity heterostructures" by Boxuan Yang, et al. The authors proposed that the magnetic proximity effect induces a spin-dependent exchange shift in the band structure of graphene, resulting in magnetization and spin polarization of electron/hole carriers. Consequently, electrostatic manipulation can be employed to exert control over the spin polarization in graphene. To validate this conclusion, van der Waals antiferromagnet CrSBr-graphene heterostructures were fabricated by the authors, and magnetotransport measurements were conducted at both low and high magnetic fields. An exchange shift ranging from 27 to 32 meV was observed, demonstrating its correlation with the spin polarization of carrier density. Furthermore, they successfully demonstrated the electrostatic modulation of spin polarization, achieving a wide range of values from +69% to -50%. These results are very important to develop the spintronic devices based on the electrostatic tunability of magnetic properties. Therefore, this work is meaningful and should be published at Nature Communications. However, some issues should be addressed in an adequate manner before I can fully support the publication of this paper.

We thank the Reviewer for recommending the publication of our work in Nature Communications. We address below the issues raised by the Reviewer.

1. The cross-sectional view of CrSBr-graphene heterostructures has not been documented in the article. Could you kindly provide a detailed the cross-sectional image of heterostructures?

A schematic cross-sectional view of our device has been provided in Fig. 1b. This image specifies the thickness of the different materials. If the referee is interested in an atomically-resolved image of the of the interface (such that can be given by a TEM cut), we do not think that such a measurement could give useful information for the following reasons.

According to DFT calculations performed by Mairbek Chshiev and Fatima Ibrahim (see response to question 1 of Reviewer 1), the magnetic proximity effect strongly depends on the van der Waals gap. A change of this gap by less than 5 Angstrom results in a one order of magnitude modification of the exchange shift. A TEM cut cannot reliably resolve such a small scale on a single interface. In addition, the cut is known to damage van der Waals materials, which would results in a modification of the bilayers structural properties. Finally, a relevant measurement of the van der Waals gap would necessitate to perform the TEM measurement on

the same device as used for the magnetotransport measurements, which is not a possibility for us.

2. Whether the magnetic proximity effect is influenced by the packing direction and thickness of the two materials, and if so, please provide relevant evidence; If not, kindly explain the underlying rationale.

A dependence of the exchange shift with the packing direction (as well as on atomic shift, layer alignment and possible twist angle) is indeed possible. However, although the crystallographic axis of CrSBr are known from the shape of the exfoliated flake, we do not know their orientation with respect to the graphene crystallographic axis. This renders it impossible to evaluate the effect of the packing direction on the exchange shift.

The purpose of our present paper is to demonstrate the presence of a electrostatically tunable spin polarization in graphene interfaced with a magnetic material. This does not necessitate to know the orientation of the materials involved in stack used in this study. Therefore, although such information would be relevant for future investigation on the graphene-CrSBr system, we believe that the influence of the packing direction on the proximity effect is outside the scope of this paper.

This point was however studied by the DFT calculations of Mairbek Chshiev and Fatima Ibrahim, in which a dependence of the exchange shift on the packing direction was observed.

Finally, the large thickness (100 nm) of CrSBr rules out any effect linked to the odd or even number of magnetic layers. Furthermore, our preliminary DFT calculations indicate that only the CrSBr layer in contact with graphene produces the magnetic proximity effect. The CrSBr thickness should therefore not play a role in the observed magnetic proximity effect.

3. The authors only present the test results of a single device, which lacks universality. Could the authors conduct experiments on multiple devices to validate their proposed theoretical model?

The measurements were conducted on the device presented in Fig. 1a. Although this constitutes a single structure, we explicitly made a Hall bar with several pairs of contacts, and a size which is much larger than the one of devices typically used in this type of experiments. We present measurements on electrically independent regions of the device, where we observed a small variation of the charge compensation gate voltage and measured exchange shift. The analysis of the magnetotransport measurement performed in these two regions using our model gave consistent results, which indicate a universality of the Landau level energy modification by the exchange shift. We think that this constitutes a strong confirmation of the universality of our model, despite the complexity of the analysis for this type of system, as detailed in the supplementary information. In addition these results give an idea of the homogeneity of the system.

Finally, the exchange shift extracted in our paper is in good agreement with the one previously extracted using non-local transport measurement on Gr-CrSBr heterostructures [Ghiasi et al, *Nat. Nanotechnol.* **16**, 788–794 (2021)]. This consistency in the results provided by different analysis is a good confirmation that our model present a consistent and universal image of the physics at play in this system.

4. The simulation results are in qualitative and quantitative agreement with their measurements of magnetic transport. However, there still exist disparities between the model and the experimental findings. Can the authors elaborate on the reasons for disparities?

There are two levels of discrepancies between our data and the results of our model. The first regards the difference in the amplitude of the measured signal with the one predicted by our model. Here, as previously done in the text, we state the basic assumptions of our model, and discuss their possible limitations for our system.

1. *The counterflowing edge channels with opposite spins do not equilibrate between the contacts.*

In contrast to the helical and spin polarized edge channels used to describe the quantum spin Hall and the quantum spin anomalous Hall effects, there exist no topological protection which prevents the scattering of a carrier into a counterpropagating state at the same edge. This could cause a partial equilibration of the edge channels in the region between the contacts, and reduce the measured transverse resistance.

2. *There is no bulk transport pathway which allows current flow other than through the edge channels.*

The dry transfer technique used for the device fabrication can create defects in the graphene flake, resulting in the formation of charge puddles in its bulk. These charge puddles may create bulk transport channels linking edge channels and decrease the measured transverse resistance with respect to the computed one. The possible presence of bulk transport pathways has been discussed in section VI of the supplementary information. The very close correspondence between the measurements and the model prediction for dataset A would tend to rule out a significant contribution of bulk conduction channels in the corresponding region of the device. Dataset B however displayed a transverse resistance amplitude smaller than the computed one. As discussed in the supplementary information (Section VI), this could indicate the presence of bulk conduction channels in the corresponding region of the device.

The second level of discrepancy between our model and the experimental results is between the predicted and measured gate dependence of the spin polarization measured at low field [Fig. 5c]. In particular, we observe that the saturation of the polarization as a function of the gate voltage occurs at a lower voltage than the one predicted by our model calculations. From our preliminary DFT calculations, it appears that the magnetic proximity effect between graphene and CrSBr does not only result in an exchange splitting, and that other modifications of the band structure might be at play. These modifications could lead to the disagreement observed between the model calculations and the measurements. We have already made a note on this possible limitation of our model in line 437 to 454 of our original manuscript:

“Although our modelling matched closely, both qualitatively and quantitatively, with our magnetotransport measurements, discrepancies between the model and the experiment remain. The most striking one is the deviation of the measured Hall coefficient from its predicted values when moving away from the charge neutrality point. Our model assumes that the only effect of

the MPE on the band structure of graphene is to cause a constant exchange energy shift, which does not depend on the carrier type or on the electronic state energy. This assumption might be too restrictive to fully account for the details of our data in view of theoretical works predicting a carrier dependent exchange shift, as well as a possible gap opening at the graphene Dirac point [44]. Furthermore, an energy dependence of the exchange shift can be expected in this type of systems [45,46]. Further refinement of our model is possible to test these theoretical predictions.”

5. The authors posit that the disparity between modeling and experimental results primarily stems from the deviation of the measured Hall coefficient from its predicted values as one moves away from the charge neutrality point. Their model assumes that the sole impact of MPE on graphene's band structure is a constant exchange energy shift, which remains independent of carrier type or electronic state energy. However, the author lacks comprehensive experimental data to substantiate the aforementioned conclusion. Can the author furnish it?

The assumption of a gate voltage independent shift of the graphene band structure as a function of the spin state is indeed the most basic assumption of the model used for the theoretical evaluation of the Hall coefficient. Due to the lack of information regarding the details of the magnetic proximity effect, this assumption is necessary to perform our calculations. The overall good agreement between our data supports this assumption.

Our preliminary DFT calculations however indicate that this is a simplification of the CrSBr proximity effect on the graphene band structure. We indeed observed the opening of a gap at the graphene Dirac point. This gap, although smaller than the calculated exchange shift, could still play some role in the magnetotransport measurement.

The assumption that the sole effect of the magnetic proximity effect in the graphene band structure is the introduction of an exchange energy shift is a first order explanation. This explanation works well since we observed an overall good agreement between our data and our model predictions. As already stated in our reply to the first remark of Reviewer 1, the conclusions of our work do not depend on the specifics of the transport properties in proximitized graphene. It is however clear that further research will be needed to fully understand this phenomena. Such a detailed study is beyond the scope of the present paper.

REVIEWER COMMENTS

Reviewer #1 (Remarks to the Author):

I think the authors have clearly addressed all the reviewers' comments. Therefore, I recommend it for publication.

Reviewer #2 (Remarks to the Author):

Although authors do not believe that additional theoretical calculations are needed, I still insist they should calculate the band spin splitting of graphene in the heterostructure composed of graphene and CrSBr through first principles calculations. This can show band spin splitting and Fermi level, explaining these results. And I think this calculation is not complicated, and the computational cost is not large. But It can supplement their experimental results.

Reviewer #3 (Remarks to the Author):

Report for "Electrostatically controlled spin polarization in Graphene-CrSBr magnetic proximity heterostructures" by Boxuan Yang, et al.

Although the authors have responded to our questions earnestly, they have not provided straightforward answers to the first two questions. Regarding the first question, the authors explained that a change of this gap by less than 5 Angstrom results in a one order of magnitude modification of the exchange shift, and TEM cut cannot reliably resolve such a small scale on a single interface. Additionally, they mentioned that cutting is known to damage van der Waals materials and would lead to modifications in bilayer structural properties. However, recent advancements in scientific research have reported the successful preparation of two-dimensional materials cross-sections using focused ion beam technique and clear observation of single-atom layer cross-sectional structures through special aberration corrected scanning transmission electron microscopy (Cs-STEM). It demonstrates the possibility of obtaining high-resolution interface images with appropriate technical means. Moreover, sample preparation quality plays a crucial role in its performance, particularly for preparing heterogeneous junctions of two-dimensional materials which remains a technical challenge. If the authors can provide a clear interface image, it will offer strong structural evidence for understanding its mechanism. The sole reliance on theoretical calculations may not provide sufficient persuasive evidence. For the second question, although the author believed that material thickness does not affect the magnetic proximity effect, it is generally accepted that thickness indeed influences various physical properties of two-dimensional materials. Therefore, the reviewer still requests that the author must provide experimental data to verify this inference, as it will enhance the integrity and credibility of the study.

Therefore, the reviewer concludes that the current version of the manuscript is not yet deemed suitable for publication in Nature Communications.

We thank the Reviewers for their remarks. We are happy to have answered almost all the remarks of the Reviewers. In our reply we address the Reviewer 2 remarks regarding the DFT calculations by pointing out that preliminary DFT results have already been provided in our previous reply to the Reviewers. Secondly, we address the proposition of Reviewers 3 to perform sub angstrom imaging of the van der Waals gap for comparison of our results with DFT calculations by noting that this is not technically possible. In addition, we stress that our conclusions do not depend on DFT calculations since the extraction of the exchange shift in our paper relies on the modification by the magnetic proximity effect of the well-established Landau level energy spacing in pristine graphene. Finally, a study on the thickness dependence of the magnetic proximity effect is in our opinion outside the scope of this paper.

Reviewer #1 (Remarks to the Author):

I think the authors have clearly addressed all the reviewers' comments. Therefore, I recommend it for publication.

We thank the Reviewer for recommending our paper for publication in Nature Communications.

Reviewer #2 (Remarks to the Author):

Although authors do not believe that additional theoretical calculations are needed, I still insist they should calculate the band spin splitting of graphene in the heterostructure composed of graphene and CrSBr through first principles calculations. This can show band spin splitting and Fermi level, explaining these results. And I think this calculation is not complicated, and the computational cost is not large. But It can supplement their experimental results.

We are a bit confused by the remark of the Reviewer. As stated in the previous reply to the Reviewers, we obtained preliminary DFT calculations of bilayer graphene proximitized with CrSBr from Mairbek Chshiev and Fatima Ibrahim (Université Grenoble-Alpes and CEA Grenoble). From the remark of Reviewer 2, we gather that the Reviewer may have overlooked this part of our reply, which was in the answer to the first remark of Reviewer 1. Here we repeat the discussion provided in the previous response to the Reviewers, which is also added with more details in the supplementary information (see Section II.D):

Although DFT calculations of bilayer graphene proximitized with CrSBr are not strictly necessary to support our claim, we have been in touch with Mairbek Chshiev and Fatima Ibrahim (Université Grenoble Alpes and CEA Grenoble) regarding the possibility to perform said calculations. This work is currently in progress, and we will summarize here the preliminary results obtained for calculations done for a bilayer graphene layer in contact with a CrSBr bilayer.

- 1. The presence of the CrSBr layer is expected to induce a hole doping in graphene, which has been observed in our measurements. We however would like to emphasize that it could originate from extrinsic doping in the actual device used for this study.*
- 2. The proximity effect induces an exchange shift which strongly depends on the van der Waals gap between graphene and CrSBr, with a modification of one order of magnitude when changing this gap by a few tenths of Angstroms.*
- 3. The exchange shift appears to depend on the relative orientation and twisting between the graphene and CrSBr crystal axis.*

We emphasize that these calculations have been performed for bilayer CrSBr on top of graphene. The CrSBr layer used in our experiments is much thicker (100 nm), such that the conclusion of these calculations should be considered with care. They however overall support our finding that the proximity effect between CrSBr and bilayer graphene can induce an exchange shift in the graphene band structure of the order of magnitude which we observed, and therefore can produce a spin polarization as discussed in our paper.

Because these calculations are still in progress, we do not report them in full, but instead summarized their preliminary conclusions in the Supplementary Information (Section II.D).

Please note that we have made a mistake in our previous response to the Reviewers, in which we stated that the approximately one order of magnitude modification of the exchange shift occurs for a few angstroms change in the van der Waals gap. The correct characteristic length scale for the modification of the exchange shift is a few *tenths of angstroms* (an order of magnitude smaller) according to the DFT calculations of Fatima Ibrahim and Mairbek Chshiev, in agreement with previous results for this type of systems [see for instance Yang et al, Phys. Rev. Lett. 110, 046603 (2013)]. We apologize for the possible confusion caused by this typo.

These preliminary theoretical calculations show that our conclusions are not in contradiction with DFT results, which do however not allow to perform a full comparison between experiment and theory at this stage. Furthermore, and as previously stated, we stress that our conclusions do not rely on DFT calculations.

Reviewer #3 (Remarks to the Author):

Report for "Electrostatically controlled spin polarization in Graphene-CrSBr magnetic proximity heterostructures" by Boxuan Yang, et al.

Although the authors have responded to our questions earnestly, they have not provided straightforward answers to the first two questions. Regarding the first question, the authors explained that a change of this gap by less than 5 Angstrom results in a one order of magnitude modification of the exchange shift, and TEM cut cannot reliably resolve such a small scale on a single interface. Additionally, they mentioned that cutting is known to damage van der Waals materials and would lead to modifications in bilayer structural properties. However, recent advancements in scientific research have reported the successful preparation of two-dimensional materials cross-sections using focused ion beam technique and clear observation of single-atom layer cross-sectional structures through special aberration corrected scanning transmission electron microscopy (Cs-STEM). It demonstrates the possibility of obtaining high-resolution interface images with appropriate technical means. Moreover, sample preparation quality plays a crucial role in its performance, particularly for preparing heterogeneous junctions of two-dimensional materials which remains a technical challenge. If the authors can provide a clear interface image, it will offer strong structural evidence for understanding its mechanism. The sole reliance on theoretical calculations may not provide sufficient persuasive evidence.

We first comment on the following sentence by Reviewer 3: "The authors explained that a change of this gap by less than 5 Angstrom results in a one order of magnitude modification of the exchange shift". From the DFT calculations provided to us by Fatima Ibrahim and Mairbek Chshiev, this statement is not correct. As stated above in the reply to the remark of Reviewer 2, a 0.5 and not 5 angstrom modification of the van der Waals gap results in a one order of magnitude modification of the exchange shift. This value reported by the Reviewer originates from a typo in our previous response, in which we

stated mistakenly that the exchange shift is modified over a 5 angstrom characteristic length. We apologize for this mistake which may have led to misunderstandings in the discussion. While the statement of the Reviewer that the van der Waals gap can be resolved over a 5 angstrom length scale using STEM measurements is correct, we argue in the following that this is not possible on a 0.5 angstrom length scale.

We now comment on the proposition by the Reviewer to perform STEM measurements to extract the van der Waals gap for comparison with DFT calculations. The Reviewer correctly points out that it is possible to make atomically resolved images of interfaces between (2D) materials. We actually did that some years ago [see Hidding et al, J. Phys. Mater. 4 (2021)04LT01 (open access)], in collaboration with our colleague Prof. Bart J. Kooi who is an expert in high resolution TEM and who was involved in making the picture in Fig 3c of the paper by Hidding et al. It shows a cross sectional HAADF-STEM image of a WSe₂/Py bilayer (obtained by 300 kV aberration corrected STEM). This was indeed made by using FIB to cut out a part of the sample. We have discussed in detail with him if it would be possible to obtain a similar picture for our case, and extract the Van der Waals gap in the required resolution between the graphene layer and CrSBr. In the opinion of Prof. Kooi, such a measurement is not possible, and he gave the following comments:

1. After FIB cutting out a slice of about 1 micrometer thickness, this has to be further thinned by ion beam polishing to about 100 nanometer which for our system would rather surely induce damage and/or stresses obscuring any proper imaging.
2. An important reason why the atomic alignment is visible in fig 3c, is due to the relatively heavy atoms of W and Se that show up with large (Z) contrast compared to the surrounding materials. Doing the same with light carbon would generate an image where the graphene would not or hardly be visible (in comparison to the CrSBr).
3. As can be seen in fig. 3c the picture of the interfaces is very irregular. This is because an extremely thin layer with a thickness of the order of a nanometer is imaged in projection in a sample that is about two orders of magnitude thicker (i.e. 100 nm). Slight bending and curvature of the interface will therefore in projection generate a spread-out (blurred) interface. This means that even in the unlikely case that it will be possible to get a reasonable picture at all, this will be the best quality picture one could hope for. It is clear that this will not allow to extract distances with an accuracy of 0.5 Å or less (one tenth of the CrSBr lattice spacing), which is needed for a comparison with theory.
4. There is an additional complication that this procedure has to be done at room temperature. Due to thermal stress, there is no guarantee that the Van der Waals gap will not change when cooling down the sample.

We conclude from these comments and opinion by an expert of high resolution TEM imagery techniques that it will be practically impossible to obtain the Van der Waals gap with sufficient accuracy on a sub angstrom scale. Furthermore, even if we could do such a measurement, it would not allow relevant conclusions of how the gap actually was in the measured device. Finally, we stress that such an additional measurement would be relevant only for comparison with DFT calculations which, as previously stated, are not necessary to support our conclusions.

For the second question, although the author believed that material thickness does not affect the magnetic proximity effect, it is generally accepted that thickness indeed influences various physical properties of two-dimensional materials. Therefore, the reviewer still requests that the author must provide experimental data to verify this inference, as it will enhance the integrity and credibility of the study.

Therefore, the reviewer concludes that the current version of the manuscript is not yet deemed suitable for publication in Nature Communications.

We do not understand which additional data the Reviewer is asking us to provide. The Reviewer is right in pointing out that the CrSBr thickness has an effect on its bulk physical properties, as previously reported in literature. It is however in our view unlikely that this would have a significant effect on the magnetic proximity effect in bilayer graphene.

Previous DFT calculations [Hallal et al, 2D Matter. 4 (2017)] have discussed the dependence of the magnetic proximity effect in graphene as a function of the magnetic material thickness. It has been found that only the first two layers have a significant effect on the exchange shift. This is in agreement with our preliminary DFT calculations which showed that the exchange shift mostly originates from the contact of bilayer graphene with the first CrSBr layer, with a second order contribution by the second CrSBr layer. It is therefore reasonable to consider that the other CrSBr layers will not have a significant effect on the magnetic proximity effect.

It is however possible that the modifications of the bulk CrSBr magnetic properties could modify the exchange shift induced by the first two layers. Such a detailed understanding of the system is however outside the scope of our paper, which aims at demonstrating the presence of a spin polarization in graphene proximitized with CrSBr. We therefore do not think that a study of the CrSBr thickness dependence of the magnetic proximity effect would modify the message of our paper.

In conclusion, although the measurements proposed by the Reviewer would indeed deepen the understanding of the magnetic proximity effect induced by CrSBr on graphene, those appear to be either not technically possible, or fall outside the scope of this paper.

REVIEWER COMMENTS

Reviewer #2 (Remarks to the Author):

Authors only summarized some preliminary conclusions from DFT calculations in the reply and Supplementary Information (Section II.D), without any data or figures available. I don't think the author's answer is honest and convincing. The atomic structures, data and figures from DFT calculations is essential to support these conclusions. These should be fully included in the supplementary materials. I think the authors have no reason to skip this section.

Reviewer #3 (Remarks to the Author):

Dear editor,

Report for "Electrostatically controlled spin polarization in Graphene-CrSBr magnetic proximity heterostructures" by Boxuan Yang, et al.

The authors have effectively addressed all the reviewers' comments, demonstrating a comprehensive response. Therefore, I would recommend accepting this manuscript at Nature Communications.

Reviewer #2 (Remarks to the Author):

Authors only summarized some preliminary conclusions from DFT calculations in the reply and Supplementary Information (Section II.D), without any data or figures available. I don't think the author's answer is honest and convincing. The atomic structures, data and figures from DFT calculations is essential to support these conclusions. These should be fully included in the supplementary materials. I think the authors have no reason to skip this section.

We are a bit surprised by the Reviewer's comment regarding the honesty of our answer. As stated in the first and second response to the Reviewers, our DFT calculations show that CrSBr induces an exchange shift in the band structure of graphene. These calculations are however in a preliminary stage, and it is not possible to use them to conclude on the agreement between theory and experiment. We did not hide this point from the reviewers.

We have followed the request by the reviewer to include figures reporting the results of our DFT calculations, and subsequently slightly adapted the corresponding discussion. Since the results of the DFT calculations are now reported in detail, we added Mairbek Chshiev and Fatima Ibrahim to the author list.

Modification of the section II.D of the supplementary information:

Preliminary density functional theory calculations of the proximity effect between graphene and CrSBr

Here we present preliminary Density Functional Theory [1] results of the band structure of a (5×1) bilayer CrSBr and (8×2) graphene supercell. The CrSBr a-axis was first aligned along the graphene armchair direction ($\theta = 0^\circ$) [Fig. 1(b)]. To explore the effect of the interfacial distance on the exchange splitting, two van der Waals gaps were considered: the relaxed one $d_z = 3.3 \text{ \AA}$ and a smaller $d_z = 2.5 \text{ \AA}$. An exchange splitting of 0.2 meV is present at the Dirac point of graphene for $d = 3.3 \text{ \AA}$ and increases up to 1 meV for $d_z = 2.5 \text{ \AA}$ [Fig. 1(a)]. This shows the extreme sensitivity of the exchange splitting on the van der Waals gap. The small value of exchange splitting results from the moderate interaction between the graphene Dirac cone lying in the $Y \rightarrow \Gamma$ path of the Brillouin zone and the bottom of the conduction band of CrSBr at the Γ point.

Figure 1: (a) Band structure of bilayer graphene/bilayer CrSBr for a van der Waals gap of 2.5 Å, with a zoom on the graphene Dirac point with the arrow showing the exchange splitted spin up (blue) and spin down (red) bands. (b) Side and top view of the supercell used in the calculation. The CrSBr a-axis is aligned along the graphene armchair direction where the stacking angle $\theta = 0^\circ$.

To enhance the interaction, we change the stacking angle between the CrSBr a-axis and the graphene armchair direction to $\theta = 30^\circ$ [Fig. 2b]. In this case, the Dirac cone folds to the Γ point. The band structure calculations of the relaxed structure with a van der Waals gap of $d_z = 3.3 \text{ \AA}$ show an exchange splitting of 7 meV [Fig. 2a]. This value is likely to increase for a reduced d_z as shown for the case with $\theta = 0^\circ$.

Figure 2: (a) Band structure of graphene/bilayer CrSBr for a van der Waals gap of 3.3 \AA , with a zoom in on the graphene Dirac point showing the exchange splitted spin up (blue) and spin down (red) bands. The black arrow shows the amplitude of the splitting. (b) Top view of the supercell used in the calculation where the stacking angle between CrSBr a-axis and the graphene armchair direction is $\theta = 30^\circ$.

It is noteworthy that for both stacking angles we found the ferromagnetic configuration of the CrSBr bilayer more stable than the antiferromagnetic configuration.

Based on those results, CrSBr induces an exchange splitting in the band structure of graphene which strongly depends on the stacking between graphene and CrSBr and also on the van der Waals gap, making the comparison between the DFT results and our experimental results uncertain as these parameters cannot be extracted for our devices. Further work will be therefore needed to understand in detail the magnetic proximity effect of CrSBr on graphene.

[1] Kresse, G. & Furthmüller, J. Efficient iterative schemes for ab initio total-energy calculations using a plane-wave basis set. *Physical Review B* 54, 11169-11186 (1996).

Reviewer #3 (Remarks to the Author):

Dear editor,

Report for "Electrostatically controlled spin polarization in Graphene-CrSBr magnetic proximity heterostructures" by Boxuan Yang, et al.

The authors have effectively addressed all the reviewers' comments, demonstrating a comprehensive response. Therefore, I would recommend accepting this manuscript at Nature Communications.

We thanks the reviewer for recommending our paper for publication in Nature communications.

REVIEWERS' COMMENTS

Reviewer #2 (Remarks to the Author):

The revised manuscript has addressed all my concerns thus is suitable to publish as is.